# Chronic Tubulointerstitial Nephropathy of Agricultural Communities

Sourabh Sharma [1],*[ID], Neha Sharma [2], Urmila Anandh [3][ID] and Swarnalata Gowrishankar [4]

1   Department of Nephrology, Vardhman Mahavir Medical College, Safdarjung Hospital, Delhi 110029, India
2   Department of Pathology, Vardhman Mahavir Medical College, Safdarjung Hospital, Delhi 110029, India; drneha.bkn@gmail.com
3   Department of Nephrology, Amrita Hospitals, Faridabad 121002, India; uanandh@gmail.com
4   Department of Pathology, Apollo Hospitals, Hyderabad 500033, India; swarnalatag@gmail.com
*   Correspondence: drsourabh05@gmail.com

**Abstract:** Chronic interstitial nephritis in agricultural communities is an emerging public health concern affecting numerous agricultural communities in tropical countries, including regions in India, with a significant impact on the health and well-being of affected individuals. The affected individuals suffer from various psychosocial, nutritional, and metabolic challenges due to organ failure, which affects their quality of life. The etiology remains poorly understood, and various risk factors, which include various environmental and occupational hazards, have been implicated in its development. The recent discovery of lysosomal proximal tubulopathy has reignited interest in its pathogenesis. Along with the representative feature of chronic interstitial nephritis, changes suggestive of tubular injury have also been reported. It is suggested to use the term "chronic tubulointerstitial nephropathy of agricultural community" instead of chronic interstitial nephritis of the agricultural communities. Chronic tubulointerstitial nephropathy in agricultural communities is a slowly progressive disease that initially does not cause any symptoms in patients and most patients have a delayed onset of symptoms. Several diagnostic criteria have been introduced over the past years and one introduced by the Ministry of Health of Sri Lanka is widely used. The management of this chronic illness is no different from other causes of chronic interstitial nephritis and our focus should be on implementing various preventive strategies to reduce its incidence in agricultural communities and protect the health and well-being of agricultural workers. By disseminating knowledge about chronic tubulointerstitial nephropathy in agricultural communities, we can contribute to the development of evidence-based interventions to reduce the burden of the disease on affected communities. Moreover, we would like to sensitize physicians to this entity to increase awareness and identify potential endemic areas in various agricultural communities.

**Keywords:** chronic interstitial nephritis; chronic interstitial nephritis of agricultural communities; preventive nephrology; lysosomal proximal tubulopathy; CKDu

## 1. Introduction

Interstitial nephritis is a kidney disease characterized by inflammation of the interstitial tissue of the kidneys. It can be acute or chronic and is often caused by an allergic reaction to medications or infections [1,2]. If left untreated, it can lead to progressive loss of kidney function and end-stage renal disease [1,2]. The main characteristics of interstitial nephritis are the presence in the interstitium of variable numbers of inflammatory cells (usually mononuclear cells, lymphocytes, and plasma cells, with smaller numbers of eosinophils in some instances and polymorphonuclear leukocytes in others), accompanied by variable degrees of tubular atrophy or loss [3]. Chronic interstitial nephritis (CIN) is a kidney disease characterized by the inflammation and scarring of the interstitial tissue of the kidneys. It is often caused by prolonged exposure to toxins or medications, autoimmune diseases, infections, or genetic factors [4,5].

Chronic interstitial nephritis of agricultural communities (CINAC) is a distinct form of chronic kidney disease (CKD) which is not associated with other common CKD etiologies such as diabetes, hypertension, and glomerulonephritis. The prevalence and incidence of CINAC have become an epidemic in several countries, including El Salvador, Guatemala, Nicaragua, and Costa Rica in Central America, Sri Lanka and India in Asia, and Egypt in Africa [6,7]. CINAC is reported by different terminology in different endemic regions such as Mesoamerican endemic nephropathy (MeN), Central American nephropathy, Salvadoran agricultural nephropathy, Uddanam endemic nephropathy in India, and Sri Lankan agricultural nephropathy in Sri Lanka or CKDu (CKD of unknown etiology). To provide a comprehensive description of the disease, the term CINAC was proposed [6,7]. As with the representative feature of chronic interstitial nephritis, changes suggestive of tubular injury have also been reported, and it is suggested to use the term "chronic tubulointerstitial nephropathy of agricultural communities (CTINAC)" instead of CINAC.

CTINAC primarily affects individuals of CTINAC-endemic areas which include persons residing or working in agriculture-based societies [7]. Environmental (hot tropical climates), socioeconomic (hardship and poverty), and occupational factors (hazardous agrochemicals through ingestion of contaminated food and water, inhalation, and occupational skin exposure) are associated with clinical features that aid in diagnosis [6,7]. The disease is more common in men, but in these agricultural communities, it also affects the pediatric and female populations. It is worth noting that in CTINAC-endemic areas, people who reside but do not work in agriculture-based occupations may also develop CTINAC [6,7]. Patients with CTINAC present with deranged renal function (CKD Stage 3/4) along with radiological evidence of bilateral shrunken kidneys with irregular contour and, specifically, absence of (or low) proteinuria. On biopsy, these cases exhibit lesions in the tubulointerstitial compartment, along with glomerulosclerosis [6].

## 2. Agricultural Communities—Problems and Challenges

Agricultural communities are groups of people whose livelihoods primarily rely on agriculture. They are often exposed to a range of environmental and occupational factors that can increase their risk of kidney diseases. Some of the key factors include exposure to pesticides, heavy metals, and other chemicals used in agriculture. These substances can accumulate in the body over time and damage the kidneys, leading to chronic kidney disease. In addition to chemical exposure, agricultural communities may also be at risk of kidney disease due to dehydration, heat stress, and other physical stresses associated with farming. Many farmers work long hours in hot, dry conditions and may not have access to adequate hydration or rest breaks, which can put a strain on their kidneys and increase their risk of kidney damage. Moreover, these communities are at an increased risk of kidney disease due to limited access to healthcare and screening services.

## 3. Epidemiology of CTINAC

In a systematic review, Liyange T et al. [8] reported highly variable prevalence estimates of CKD (7.0% to 34.3%). They reiterated that estimated that around 434 million people were suffering from CKD across southern, eastern, and south-eastern Asia [9–15]. There have been multiple reports of clusters with a high CKD prevalence in multiple regions of Central America and South Asia, particularly among agricultural communities [7,8].

The gravity of the situation can be estimated from the report of the Pan American Health Organization which stated that El Salvador and Nicaragua had the highest mortality rate related to CKD, with rates of 416 and 428 deaths per million population, respectively, which is around four times higher than any other country [16]. In El Salvador, CKD is the second most common cause of death for young people, affecting men three times more frequently than women. CTINAC comprises one of the leading causes of CKD in these CTINAC-endemic areas [7]. In these farming communities, women, men, and even children suffer, even if they do not work on the farms [17].

If we go through available literature, CTINAC still remains a major public health challenge in Sri Lanka's North Central Province even two decades after reporting of the first such case. The disease is estimated to affect over 60,000 people annually and is responsible for more than 20,000 deaths [6,18,19]. Over the period, nearby agricultural areas in northwestern, eastern, northern, and Uva provinces have been affected, covering almost a third of this tropical country. On the other hand, there have been very few reported cases from the northern province of Sri Lanka, which has the same climate, water, land, agricultural practices, and employment trends as other endemic regions. According to hospital statistics, the number of CTINAC cases increased steadily between 2000 and 2015 [6,20], and 82% of CKD patients examined by Athuraliya TN et al. [6,21] had no known underlying cause. In the North Central Province of Sri Lanka, around two-thirds of CKD patients had an unknown cause. A significant proportion of patients with unknown etiology were already in stage 4 (40%) at presentation, with 31.8% and 4.5% in stages 3 and 5, respectively. Patients in stages 1 and 2 accounted for only 3.4% [22].

India's contribution to the global CKDu burden is estimated at 16%, although data on the condition in India is limited [23,24]. Uddanam is a verdant region in the Srikakulam district of Andhra Pradesh state, India, famous for its abundant coconut and cashew plantations. An undetermined number of residents in this area suffer from CKDu [25,26], which was named Uddanam nephropathy during the International Society of Nephrology-World Congress of Nephrology 2013 in Hong Kong [25,27]. A recent report by Tatapudi R R et al. [14] on the rural population of Uddanam, Andhra Pradesh, India, revealed a CKD prevalence of 18.23%, 4 to 18 times higher than in other studies [23]. Strikingly, traditional risk factors such as persistent hypertension, diabetes, and significant proteinuria were absent in 73% of patients. Furthermore, unpublished cross-sectional estimates suggest that the prevalence of CKDu in Uddanam ranges from 40% to 60% (T Raviraju, Dr, NTR University of health sciences, personal communication with Gadde P et al. [25], August 2017), which is nearly three times higher than the national prevalence of 17.2% [24–26]. As of 2015, the number of deaths from chronic kidney disease in the last ten years was estimated to be over 4500, with around 34,000 people having kidney diseases in Uddanam [24,25,28]. One of the authors (Sharma S) of this article had encountered many cases of CKD with an unknown etiology while practicing in Sriganganagar, Rajasthan, where he used to deal with patients from the Malwa region of Punjab. We strongly believe that these cases also come under this category of CTINAC as most cases were male agricultural workers (sugarcane and paddy farms) in the 30–40 year age group, working in harsh environmental conditions.

## 4. Risk Factors for Developing Chronic Interstitial Nephritis in Agricultural Workers

The various risk factors that have been implicated in CTINAC are depicted in Table 1. Heat stress and dehydration have long been implicated as a cause of CTINAC in endemic areas [29,30].

**Table 1.** Risk factors for developing CTINAC.

| Risk Factors | Comments |
| --- | --- |
| Occupation | ❖ Farmers (paddy farmers in South Asia and sugarcane farmers in Central America [6])<br>❖ Agricultural workers |
| Age | ❖ Age above 60 years (Athuraliya NT et al. [29])<br>❖ Above 39 years (WHO [6]) |
| Sex | ❖ Male sex |
| Genetic | ❖ Single nucleotide polymorphism in SLC13A3 (PAF 50%; OR~2.13) [31] |
| Heat stress | ❖ Contributing factor<br>❖ Hot and humid climate [32] |

**Table 1.** *Cont.*

| Risk Factors | Comments |
| --- | --- |
| Recurrent dehydration | ❖ Contributing factor<br>❖ consumption of less than 3 L of water per day [32]<br>❖ Subclinical rhabdomyolysis<br>❖ Hyperuricemia and hyperuricosuria<br>❖ Aldose reductase-fructokinase pathway activation (secondary to hyperosmolarity)<br>❖ Vasopressin |
| Water source | ❖ Well water consumption<br>❖ Heavy metals (Arsenic, Lead, Cadmium, Mercury)<br>❖ Agrochemicals |
| Food source | ❖ Contamination with agrochemicals<br>❖ Contamination with microbial toxins [33,34] |
| Extreme physical exertion | ❖ Profuse sweating during work<br>❖ Working > 6 h a day in field standing in sun [6,32] |
| Nephrotoxic drugs | ❖ NSAIDs<br>❖ Aristolochic acid<br>❖ Traditional remedies |
| Infections | ❖ Malaria<br>❖ Hantavirus [35] |
| High fructose drinks | ❖ Fructose-containing beverages such as sugarcane juice<br>❖ Especially if consumed after prolonged dehydration |
| Snake bite | |
| Family history of CKD | ❖ First-degree relatives [23] |
| Neighbor with CKD | ❖ At least one neighbor suffering from CKD [23] |

In 2010, Torres C et al. [30] discovered while studying Mesoamerican nephropathy that recurrent dehydration played a role in the development of CTINAC. They proposed several potential mechanisms for this pathogenesis, including hyperuricemia and hyperuricosuria, subclinical rhabdomyolysis, aldose reductase-fructokinase pathway activation secondary to hyperosmolarity, and vasopressin [6,36–38]. Jimenez CAR et al. [39] suggested that dehydration may cause renal injury by activating the polyol pathway, which is injurious to the kidneys due to fructose generation and fructokinase metabolism of this endogenously produced fructose which is ultimately metabolized to inflammatory molecules, nephrotoxic oxidants, and urate. Fructokinase is mainly expressed in proximal convoluted tubules, which are mainly responsible for fructose metabolism. Recurrent dehydration leads to recurrent aldose reductase stimulation, causing recurrent fructose generation in proximal convoluted tubules and ultimately irreversible interstitial inflammation and tubular injury. Fructose-rich beverages such as sugarcane juice are often consumed by agricultural workers, and it could be potentiating this damage. It has been proved in various animal models that adequate fluid intake during dehydration protects the kidneys [39]. However, there are numerous disputes regarding the implication of recurrent dehydration or heat stress as the main etiological factor [6,40–42]. Based on these disputes, it had been postulated that they may be one of the contributing factors and not etiological factors for the development of CTINAC. In 2016, during World Health Organization International Consultation Workshop on CKDu in Sri Lanka, the expert committee reiterated that heat stress or dehydration is not the major risk factor but an important contributing factor to the CTINAC pathogenesis [6].

In 2007, Wanigasuriya KP et al. identified various risk factors for CTINAC in Sri Lanka, including physical labor on farms, pesticides, consuming well water, family history of kidney dysfunction, receiving indigenous traditional remedies, and experiencing a past snake bite [43]. In 2011, another study found that the risk factors for CTINAC were aged over 60 years,

working as a farmer, having a family history of CKD, and exposure to agrochemicals [29]. No significant association was found with analgesic use. An additional study suggested that cadmium (Cd) contamination of food could be a potential cause of disease [44]. They found raised Cd levels in rice, tobacco, and lotus rhizome. In an international study involving twelve countries, rice was compared, and Sri Lankan rice was found to have high Cd content, with only Bangladesh rice having a higher content [45]. According to the WHO research group, individuals over thirty-nine years old and those working in vegetable cultivation had an increased risk of developing CTINAC [6]. They also discovered that around one-third of CTINAC cases had pesticide residues above reference levels in their urine [6]. In addition, the WHO study group found that urinary Cd levels had a clinically significant correlation with the stage of CKD, indicating a dose-effect relationship [6].

In 2014, Nanayakkara S, et al. [31] conducted a genome-wide association study and identified genetic susceptibility as a risk factor for CTINAC. The study found a significant association between CTINAC and a single nucleotide polymorphism (SNP) in SLC13A3 (sodium-dependent dicarboxylate transporter member 3). The SNP had an odds ratio of 2.13, and the population-attributable fraction was 50% [6,31].

Several toxic compounds present in groundwater and soil of CTINAC-endemic agricultural regions have been implicated as possible etiological factors for CTINAC. In 2011, Chandrajith R et al. postulated that raised fluoride concentration in groundwater could be associated with an increased CTINAC prevalence in various regions of Sri Lanka [46]. Another recent study found a high number of CTINAC patients in areas where soil vanadium concentrations are high [47]. Ochratoxin A, a fungal toxic molecule, was also suggested to be a possible cause of CTINAC in Sri Lanka [33]. Additionally, the cyanobacterial toxin has also been implicated as a possible etiological factor in various CTINAC-endemic areas [34].

## 5. Pathophysiology and Pathology of CTINAC

The CTINAC epidemic is defined as a rise in the incidence of CKD cases in agricultural communities, where three key factors prevail: poverty and its associated consequences, unfavorable working conditions, and an environment contaminated with toxins [48,49]. These factors connect the disease to underlying socioeconomic circumstances that are deeply rooted [49]. The etiology of CTINAC can be broadly categorized into a triad of agrochemicals, heat stress and dehydration, and natural contaminants. This has been discussed widely under risk factors. Figure 1 depicts the etiological triad of CTINAC.

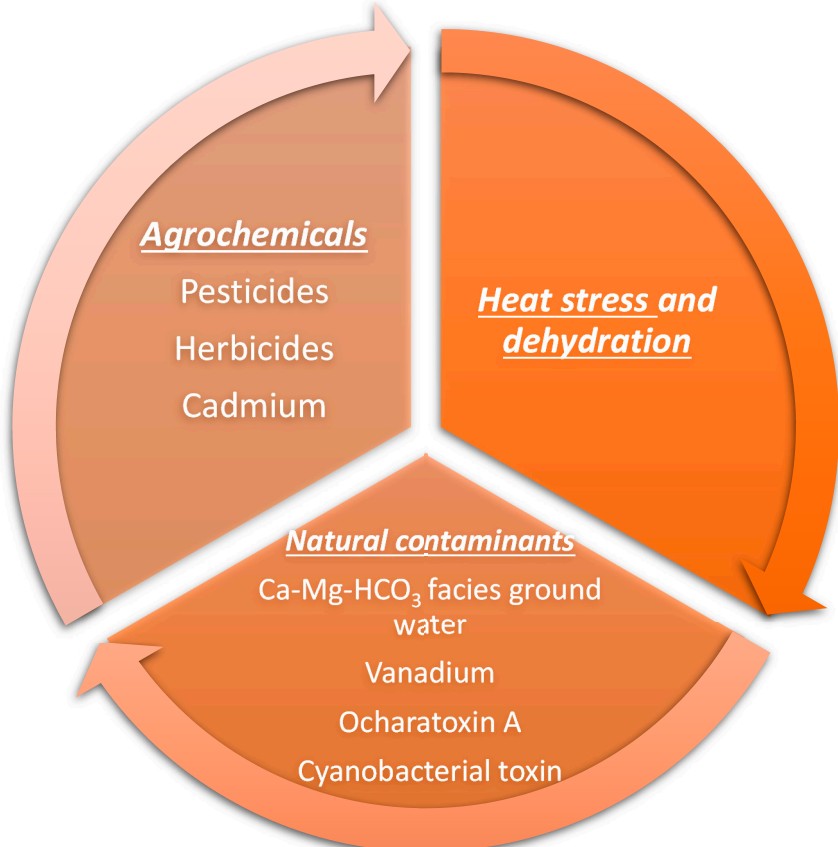

**Figure 1.** Etiological triad of CTINAC [29,32–39,43–51].

The primary histological characteristic of CTINAC is chronic interstitial nephritis, resulting from exposure to environmental toxins [52,53]. Various non-specific features have been reported which include interstitial fibrosis, tubular atrophy, glomerulosclerosis, and vascular sclerosis [52,53]. The reports of CKD patients with similar clinicopathological characteristics in various agricultural communities worldwide led to a postulation that a common subset of nephrotoxic molecules or a commonly shared pathway is responsible for causing renal injuries. The common injurious molecule shared by agricultural communities, agrochemicals, have been implicated in this disease, and hence the condition has been termed chronic agrochemical nephropathy [54]. In addition, reports have surfaced about the identification of proximal tubulopathy with enlarged dysmorphic lysosomes, accompanied by varying degrees of cell fragment shedding, epithelial atrophy, and limited proliferative capacity of proximal convoluted tubules, raising further interest in the disorder [55]. In their recent publication, Rodrigo C et al. [56] have implicated oxidative stress to be the final common pathway in the pathogenesis of CTINAC. Figure 2 depicts the detailed pathogenesis of CTINAC based on this pathway.

Oxidative stress arises from an imbalance between oxidative free radical production and the antioxidant defenses, which can be endogenously produced or derived from dietary sources. If the latter is overwhelmed, there will be an excess of peroxides and free radicals that can cause lipid peroxidation, protein oxidation, and nucleic acid damage. These effects can ultimately lead to interstitial inflammation, fibrosis, and various inflammatory syndromes [56–58]. Proximal tubular cells are especially prone to oxidative stress because they generate reactive oxygen species (ROS) during ATP production. Various animal models have shown that proximal convoluted tubules cannot produce glutathione and rely on circulating glutathione as a defense against ROS [56,59]. Intriguingly, the pathology of CTINAC tends to involve proximal tubules.

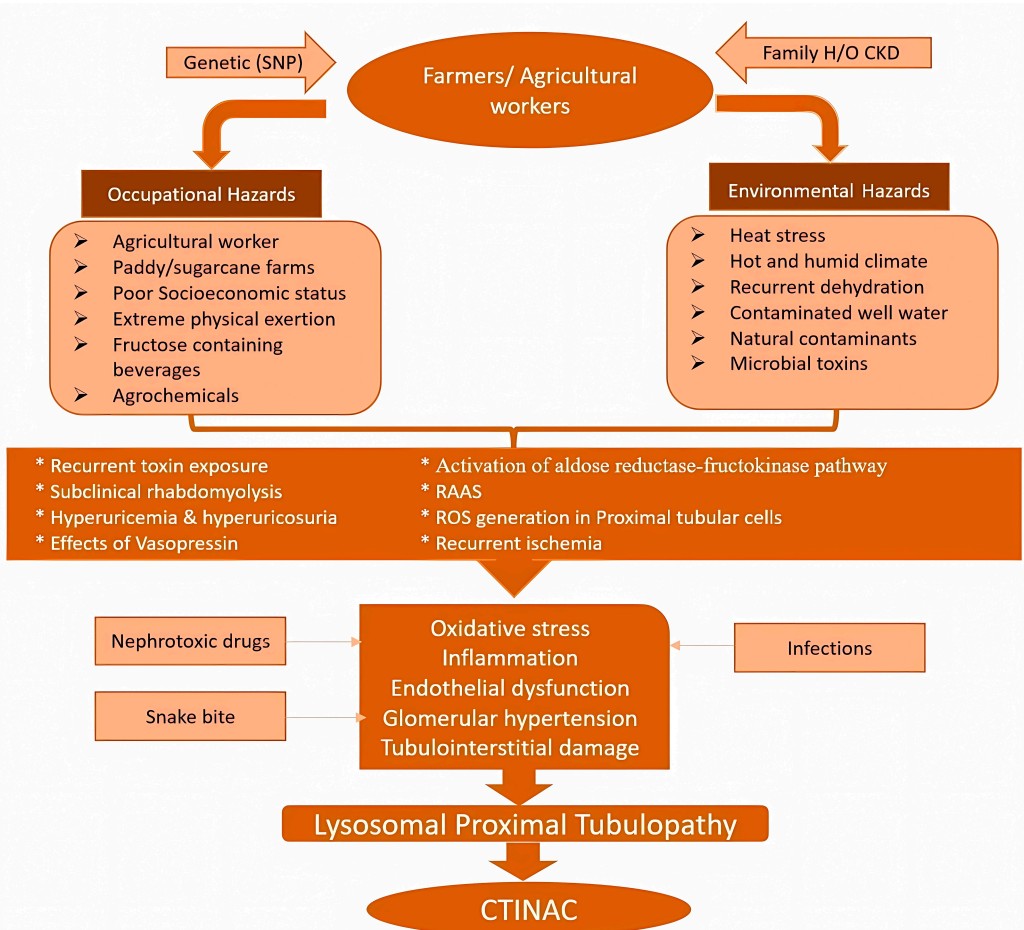

**Figure 2.** Pathogenesis of CTINAC.

The lysosomal proximal tubulopathy [56,60] has been suggested as a sine qua non of CTINAC and has been proven to be an inclusion body tubulopathy. Intriguingly, calcineurin inhibitor toxicity also produces similar lysosomal lesions. Several drug/toxin-induced nephropathies and a subgroup of patients with light chain tubulopathy also exhibit similar lesions [56,60].

One of the authors (Gowrishankar S) has been performing a histopathological evaluation of patients with suspected CTINAC in the Uddanam region of India for many years. The kidney biopsies stay instrumental in diagnosing such cases and labeling them Uddanam nephropathy. Figure 3 depicts biopsy findings in such cases with varying degrees of interstitial inflammation with little (ischemic) or no involvement of the glomerular and vascular compartments. In different instances of Uddanam nephropathy, there was a notable similarity in histopathology, where the glomeruli that were still functional appeared normal and showed no indications of increased cellularity, segmental lesions, or crescents. There was varying evidence of ischemic changes and glomerular obsolescence. Rarely, glomerular collapse and varied sclerosis along with fibrous intimal thickening and arteriolar hyalinosis have been documented in various globally reported regional nephropathies [6,26]. The primary observation was tubular atrophy and interstitial fibrosis. The inflammation was diverse, mostly lymphomononuclear, and found surrounding the atrophic tubules [61]. Vascular changes were generally absent. Direct immunofluorescence did not reveal any significant deposits in the glomeruli or along the tubular basement membrane [61]. The presence of these lesions in biopsies from individuals with low or no proteinuria living in agricultural environments confirms the diagnosis of CTINAC and highlights the importance of performing biopsies for diagnostic, prognostic, and potential therapeutic purposes [62].

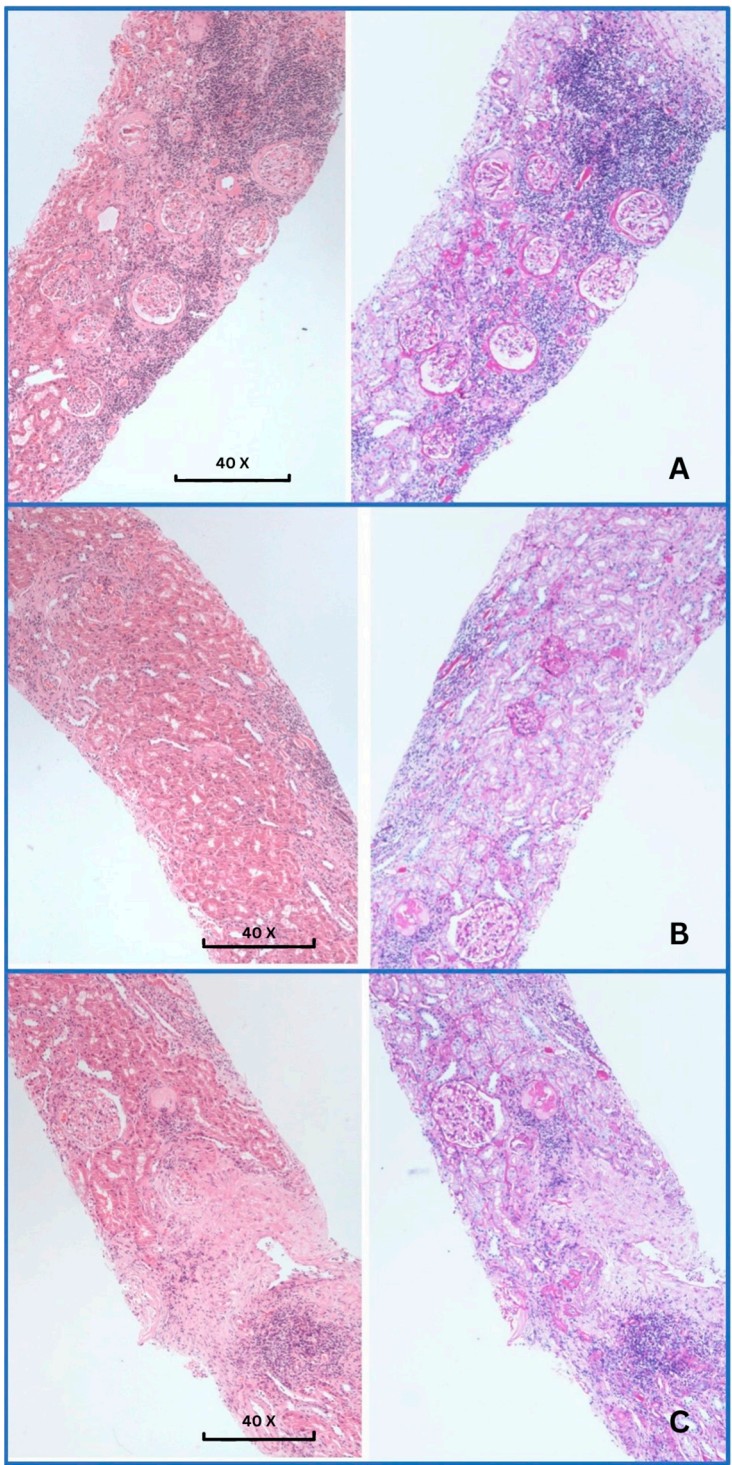

**Figure 3.** Histopathological findings in case of Uddanam nephropathy: (**A**) Glomeruli with ischemic changes and dense interstitial inflammation (40×, H&E left and PAS right); (**B**) Normal glomeruli and tubules with patchy mild interstitial inflammation (40×, H&E left and PAS right); (**C**) Focus of acellular fibrosis, surrounded by inflammation. Normal vessels (40×, H&E left and PAS right).

## 6. Clinical Presentation and Diagnosis

CTINAC is a slowly progressive disease that initially does not cause any symptoms in patients [63]. According to Athuraliya et al. [39], the symptom profile of CKDu patients was mostly normal except for enlarged serum creatinine levels (mean serum creatinine level 2.51 ± 2.06 mg/dL) and shrunken kidneys with increased echogenicity (mean relative bipo-

lar length, 5.36 ± 0.91 cm). Only a few urine samples had hyaline or granular casts, while most of them had no active sediment. The disease is characterized by tubular proteinuria (α1 and β2 macroglobulin) and elevated levels of urine neutrophil gelatinase-associated lipocalin (NGAL) (>300 ng/mg creatinine/dL) [39,64]. An early-stage increase in the urinary excretion of α1 macroglobulin was observed in CTINAC patients by Nanayakkara et al. [65], proving that tubular dysfunction and tubular proteinuria develop in CTINAC at an early stage. Furthermore, Wijkstrom et al. [53] reported that CTINAC patients have higher urinary magnesium levels.

Several diagnostic criteria have been introduced over the past years for CTINAC. In 2009, the Ministry of Health of Sri Lanka [66] introduced a diagnostic criterion to differentiate CTINAC from usual CKD:

(A) No history of diabetes, hypertension, glomerulonephritis, snakebite, or urologic disease of known etiology;

(B) Normal HbA1c level;

(C) BP < 160/100 mm Hg untreated or BP < 140/90 mm Hg on two or fewer antihypertensive medications.

Over a period, different diagnostic modalities have been introduced which were published by Wanigasuriya et al. [67], Athuraliya et al. [39], Ratnayake et al. [68], and De Silva et al. [69]. Recently, the Sri Lanka Society of Nephrologists established a diagnostic criterion for CTINAC which is most relevant to the present scenario [70]. In this criterion, cases are diagnosed as either suspected, probable, or confirmed CTINAC/CKDu:

1 Suspected CTINAC-

 a Essential criteria: CKD stage 3–5 (eGFR calculation by CKD-EPI equation)/ albuminuria ≥ 30 mg per gram creatinine/proteinuria ≥ 150 mg per gram creatinine.

 b Exclusion criteria: Proteinuria > 3000 mg per gram creatinine/albuminuria > 300 mg per gram creatinine/diabetes/hypertension (on two or more antihypertensive medications or untreated BP > 160/100 mm Hg)/history of dialysis requiring acute kidney injury/age > 70 years.

2 Probable CTINAC-

 a Essential criteria: Criteria of suspected CTINAC persistent on repeat evaluation after 3 months.

 b Exclusion criteria: Diabetes mellitus/clinicopathological evidence of the presence of other known CKD etiologies/radiological evidence of unequal kidney sizes with a discrepancy of >15 mm, obstructive nephropathy, kidney stones with either obstruction or non-obstructive single stone > 1 cm/non-obstructive unilateral or bilateral multiple stones > 5 mm in size.

3 Confirmed CTINAC-

Criteria of probable CTINAC with positive findings on histopathological examination (To be assumed as a confirmed case even in the absence of histopathology if there is any contraindication for kidney biopsy).

## 7. Prevention and Management

To prevent CTINAC, the following preventive strategies can be implemented:

1. Education and training: educating agricultural workers about the risks associated with exposure to nephrotoxic chemicals and training them on the safe handling, storage, and use of these chemicals is essential.

2. Safe food: we need to ensure the delivery of safe food to agricultural workers to avoid recurrent toxin exposure.

3. Safe drinking water: We need to monitor and ensure the delivery of toxin-free safe drinking water in CTINAC-endemic areas. This will require political will and the involvement of community workers. We need to teach agricultural workers about adequate fluid intake and avoid fructose-rich fluids.

4.  Healthy workplaces: farm workers should be employed in a healthy work environment with frequent breaks, drinking water facilities, healthy food, and fixed working hours.
5.  Use of protective equipment: Providing agricultural workers with personal protective equipment such as gloves, boots, and masks can help prevent exposure to nephrotoxic chemicals.
6.  Monitoring exposure: Regular monitoring of agricultural workers for exposure to nephrotoxic chemicals can help identify individuals who are at risk of developing CTINAC and allow for early intervention.
7.  Regulation and enforcement: Governments and regulatory bodies should implement and enforce strict regulations on the use of nephrotoxic chemicals in agriculture to protect workers from exposure.
8.  Safe disposal: Proper disposal of nephrotoxic chemicals and their containers is essential to prevent environmental contamination and reduce the risk of exposure to agricultural workers.
9.  Alternatives to nephrotoxic chemicals: Encouraging the use of safer, non-nephrotoxic chemicals in agriculture or adopting alternative methods of pest control can reduce the risk of exposure to nephrotoxic chemicals.
10. Regular health check-ups: Regular health check-ups of agricultural workers can help detect the early signs of CTINAC and allow for timely intervention.
11. Research: we need to promote collaborative basic and clinical research for a better understanding of CTINAC pathophysiology and identify protective and preventive strategies against CTINAC.

By implementing these preventive strategies, it is possible to reduce the incidence of CTINAC in agricultural communities and protect the health and well-being of agricultural workers. These strategies have been summarized in Figure 4.

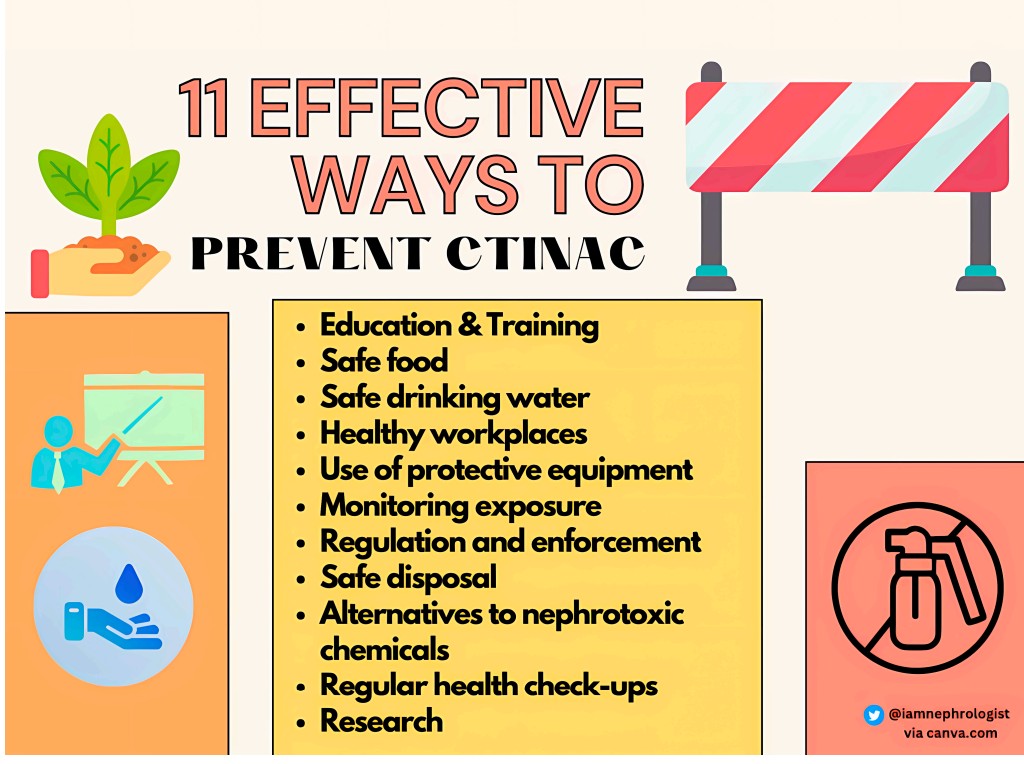

**Figure 4.** Preventive strategies for CTINAC.

The medical management of CTINAC is like other cases of chronic interstitial nephritis. It involves several key strategies aimed at slowing the progression of the disease and minimizing complications. These strategies include regular physical activity, controlling blood

pressure, reducing dietary salt intake, managing metabolic acidosis, anemia, and metabolic bone disease and other supportive treatment. Early identification and management of CTINAC are critical, and close collaboration between healthcare providers and patients is necessary to achieve optimal outcomes. Ongoing research into the underlying causes of CTINAC is also essential for developing more targeted and effective treatments.

## 8. Recent Advances and Future Research in the Understanding of CTINAC

The most important advancement in the field of CTINAC pathophysiology has been the discovery of lysosomal proximal tubulopathy [56]. It has already been discussed in detail wide supra. This has been one of the most important steps toward a better understanding of CTINAC pathogenesis.

Rodrigo C et al. [56] have postulated oxidative stress as a common final pathway of CTINAC pathogenesis which already has been discussed. In their article published in 2021, they have discussed in detail the possible role of antioxidants in the management of CTINAC to delay the progression. They have related the animal study of antioxidants against lysosomal membrane lipid peroxidation [71] with CTINAC. The role of antioxidants in delaying the progression of CKD had already been shown in a systematic review in 2012 [72]. They reported that antioxidants significantly lowered serum creatinine levels, improved creatinine clearance, and slowed the progression of disease in pre-dialysis CKD patients.

Recently, in their article, Vervaet BA et al. [62] stressed addressing new research avenues in order to progress on etiology, mechanism, treatment, prevention, and general awareness. In order to address this issue, continued research is very important and will need collaboration between researchers, the government, various farmer organizations, and the World Health Organisation. We hereby suggest a few directions for research in CTINAC:

1.  Further epidemiological studies investigate the prevalence and incidence of chronic interstitial nephritis in agricultural communities in various regions and countries and identify risk factors associated with the disease.
2.  Development and validation of non-invasive diagnostic biomarkers for early detection of CTINAC, including urinary and blood-based markers.
3.  Investigation of the mechanisms by which environmental toxins, such as pesticides and heavy metals, cause CTINAC, and development of effective prevention and intervention strategies.
4.  Evaluation of the impact of CTINAC on the overall health and well-being of affected individuals, including physical, social, and economic consequences.
5.  Examination of potential interactions between CTINAC and other health conditions prevalent in agricultural communities, such as infectious diseases and malnutrition.
6.  To Investigate potential genetic and epigenetic factors contributing to the development of CTINAC.
7.  Exploration of novel therapeutic approaches for the treatment of CTINAC, including targeted drug therapies and regenerative medicine approaches.
8.  Development and implementation of educational programs aimed at raising awareness of chronic interstitial nephritis among agricultural workers and their families, as well as healthcare professionals and policymakers.
9.  Examination of potential links between CTINAC and environmental sustainability, including impacts on agricultural productivity and food security.
10. Collaboration across disciplines and sectors to address the complex and multifaceted nature of CTINAC, including partnerships with government agencies, non-governmental organizations, and community-based organizations.

## 9. Conclusions

CTINAC is a serious health issue that affects many agricultural workers worldwide. Long-term exposure to pesticides, heavy metals, and other harmful chemicals, coupled with poor working conditions and inadequate access to healthcare, can increase the risk of developing CTINAC. Despite the growing awareness of the disease, much more research

needs to be carried out to prevent and manage its occurrence. For instance, governments and policymakers need to enforce strict regulations on the use of pesticides and other harmful chemicals and ensure that farmers and other agricultural workers have access to protective gear and safe working conditions. In addition, healthcare providers must be educated on the symptoms and risk factors of CTINAC, and there must be a concerted effort to improve access to early detection and treatment services, as well as long-term care and support for affected individuals. There are many CTINAC-endemic areas which still need to be identified and focused upon for implementing various preventive strategies. Further etiological, pathogenesis, and interventional research is needed to reduce preventable regional risk factors of CTINAC. By working together, we can raise awareness about CTINAC, promote prevention and early detection, and improve the overall health and well-being of agricultural communities around the world.

**Author Contributions:** Conceptualization, U.A. and S.S.; Methodology, U.A. and S.S.; Validation, U.A. and S.S.; Formal Analysis, U.A. and S.S.; Resources, U.A. and S.S.; Writing—Original Draft Preparation, U.A., S.S., S.G. and N.S.; Writing—Review and Editing, U.A. and S.S.; Visualization, U.A. and S.S.; Supervision, U.A.; Project Administration, U.A. All authors have read and agreed to the published version of the manuscript.

**Funding:** This research received no external funding.

**Institutional Review Board Statement:** Not applicable.

**Informed Consent Statement:** Informed consent was obtained from all subjects involved in the study.

**Data Availability Statement:** No new data was created in this review article.

**Conflicts of Interest:** The authors declare no conflict of interest.

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
