# Peer review of "Chronic Tubulointerstitial Nephropathy of Agricultural Communities"

_kidneydial, doi:10.3390/kidneydial3020021_

Round 1

Reviewer 1 Report

This review has summarized the main results of study on CKD of uncertain etiology(CKDu) in Asia including Sri Lanka,India and Central America including El Salvador, Guatemala, Nicaragua, and Costa Rica. The epidemiology, risk factors, histopathology and possible pathophysiology of CKDu has been described systemically. It was pointed that chronic interstitial nephritis was the representative feature of CKDu, often affected peasants of agricultural community, the name of chronic interstitial nephritis of agricultural community(CINAC) has been proposed. However, other features such as acute tubular injury, which was characterized by loss of brush border and cellular fragment shedding of proximal tubules, along with enlarged dysmorphic lysosomes identified by electron microscopy, has been reported, so chronic interstitial nephritis was not the only histopathological characteristics of CKDu, it is suggested to use chronic tubulointerstitial nephropathy of agricultural community or CKD of agricultural community, to replace CINAC.

Furthermore, some of reports on CKDu has found varied glomerular sclerosis with ischemic wrinkled capillary loops and dilation of Bowman's capsule, some glomeruli showed compensatory enlargement, and vascular lesions has not been paid attention in majority reports, very a few reports has also described ischemic glomerulosclerosis with arteriolar hyalinosis. These histopathological features should also added in the manuscript.

Author Response

Response to Reviewers

Q1. This review has summarized the main results of study on CKD of uncertain etiology(CKDu) in Asia including Sri Lanka,India and Central America including El Salvador, Guatemala, Nicaragua, and Costa Rica. The epidemiology, risk factors, histopathology and possible pathophysiology of CKDu has been described systemically. It was pointed that chronic interstitial nephritis was the representative feature of CKDu, often affected peasants of agricultural community, the name of chronic interstitial nephritis of agricultural community(CINAC) has been proposed. However, other features such as acute tubular injury, which was characterized by loss of brush border and cellular fragment shedding of proximal tubules, along with enlarged dysmorphic lysosomes identified by electron microscopy, has been reported, so chronic interstitial nephritis was not the only histopathological characteristics of CKDu, it is suggested to use chronic tubulointerstitial nephropathy of agricultural community or CKD of agricultural community, to replace CINAC.

Ans: Respected Sir/madam, thank you for reviewing our manuscript and for the valuable suggestions. As per your suggestions, we have changed the terminology from “chronic interstitial nephritis of agricultural community” to “chronic tubulointerstitial nephropathy of agricultural community.” [Page 1, Line 19-22; Page 2, Line 57-60]

Q2. Furthermore, some of reports on CKDu has found varied glomerular sclerosis with ischemic wrinkled capillary loops and dilation of Bowman's capsule, some glomeruli showed compensatory enlargement, and vascular lesions has not been paid attention in majority reports, very a few reports has also described ischemic glomerulosclerosis with arteriolar hyalinosis. These histopathological features should also added in the manuscript.

Ans: Respected sir/madam, We have included the glomerular and vascular changes in the manuscript. [Page 2, Line 71-72; Page 8, Line 242-244]

Reviewer 2 Report

1. The phrase "with a significant impact on the health and well-being of affected individuals" requires clarification.

 2. Change "introduced over past" to "introduced over the past."

 3. Change "increase awareness and identify" to "increase awareness and to identify."

 4. Change "men and even children suffer, even if they" to "men and even children suffer, even if they do not work in the farms."

 5. Change "CINAC still stays major" to "CINAC still remains a major."

 6. Change "affect over 60,000 people annually" to "affects over 60,000 people annually."

 7. Change "get affected" to "have been affected."

 8. Change "potential genetic and epigenetic factors that contribute to the development of CINAC" to "potential genetic and epigenetic factors that contribute to the development of CINAC."

 9. Change "world health organisation" to "World Health Organization" (capitalization error).

1. The phrase "with a significant impact on the health and well-being of affected individuals" requires clarification.

 2. Change "introduced over past" to "introduced over the past."

 3. Change "increase awareness and identify" to "increase awareness and to identify."

 4. Change "men and even children suffer, even if they" to "men and even children suffer, even if they do not work in the farms."

 5. Change "CINAC still stays major" to "CINAC still remains a major."

 6. Change "affect over 60,000 people annually" to "affects over 60,000 people annually."

 7. Change "get affected" to "have been affected."

 8. Change "potential genetic and epigenetic factors that contribute to the development of CINAC" to "potential genetic and epigenetic factors that contribute to the development of CINAC."

 9. Change "world health organisation" to "World Health Organization" (capitalization error).

Author Response

Response to Reviewers

Q1. The phrase "with a significant impact on the health and well-being of affected individuals" requires clarification.

Ans: Respected Sir/madam, thank you for reviewing our manuscript and for the valuable suggestions. As per your suggestions, we have clarified the phrase. [Page 1, Line 14-16]

Q2. Change "introduced over past" to "introduced over the past."

 Ans: Respected Sir/madam, we have incorporated these changes.

Q3. Change "increase awareness and identify" to "increase awareness and to identify."

Ans: Respected Sir/madam, we have incorporated these changes.

Q4. Change "men and even children suffer, even if they" to "men and even children suffer, even if they do not work in the farms."

Ans: Respected Sir/madam, we have incorporated these changes.

Q5. Change "CINAC still stays major" to "CINAC still remains a major."

 Ans: Respected Sir/madam, we have incorporated these changes.

Q6. Change "affect over 60,000 people annually" to "affects over 60,000 people annually."

Ans: Respected Sir/madam, we have incorporated these changes.

Q7. Change "get affected" to "have been affected."

Ans: Respected Sir/madam, we have incorporated these changes.

Q8. Change "potential genetic and epigenetic factors that contribute to the development of CINAC" to "potential genetic and epigenetic factors that contribute to the development of CINAC."

 Ans: Respected Sir/madam, we have incorporated these changes.

Q9. Change "world health organisation" to "World Health Organization" (capitalization error).

Ans: Respected Sir/madam, we have incorporated these changes.